# Heavy Metal Contamination in an Industrially Affected River Catchment Basin: Assessment, Effects, and Mitigation

**DOI:** 10.3390/ijerph18062881

**Published:** 2021-03-11

**Authors:** Gor Gevorgyan, Armine Mamyan, Tatevik Boshyan, Tigran Vardanyan, Ashok Vaseashta

**Affiliations:** 1Scientific Center of Zoology and Hydroecology, National Academy of Sciences of Republic of Armenia, Yerevan 0014, Armenia; a_mamyan@mail.ru (A.M.); tatevik-b87@mail.ru (T.B.); vardtigran@mail.ru (T.V.); 2Office of Applied Research, International Clean Water Institute, Manassas, VA 20112, USA; prof.vaseashta@ieee.org; 3Biomedical Engineering and Nanotechnologies Institute, Riga Technical University, 1658 Riga, Latvia

**Keywords:** heavy metal, contamination, catchment basin, assessment, mitigation, public health

## Abstract

The concentrations of some heavy metals (Fe, Zn, Mn, Cu, Mo, Pb, Cd) were measured in river waters, macrozoobenthos, and fish (Kura scrapers) from one of the most developed mining areas in Armenia, the Debed River catchment basin. In order to assess heavy metal contamination and its hydro-ecological and health effects, the macrozoobenthos quantitative and qualitative parameters, geo-accumulation index, and hazard index were determined. Microalgal extraction experiments were conducted to assess the microalgal remediation efficiency for heavy metal removal from mining wastewaters. The results showed that the rivers in many sites were polluted with different heavy metals induced by mining activities, which adversely affected macrozoobenthos growth and caused human health risks in the case of waters used for drinking purposes. However, the river fish, particularly Kura scrapers, were determined to be safe for consumption by the local people, as per the conditions of the evaluated fish ingestion rate. The results have shown that microalgal remediation, particularly with *Desmodesmus abundans* M3456, can be used for the efficient removal ~(62–100%) of certain emerging contaminants (Mn, Pb, Cu, Zn, Cd) from mining wastewater discharged in the Debed catchment basin.

## 1. Introduction

Rivers are exposed to severe contamination due to the rapid development of society and economy [1]. The rapid development of industrialization, in terms of increased discharges of untreated or inadequately treated wastewaters, has become a serious threat to rivers [2]. These contaminants mainly refer to heavy metals (HMs) that are persistent, tend to bio-accumulate, and are toxic substances [3,4,5,6]. Metals enter riverine environments from numerous natural and anthropogenic sources, such as geologic weathering, atmospheric precipitation, and agricultural, domestic, and industrial wastes [5]. HM pollution is causing stresses on rivers worldwide, especially in developing countries [4]. They are environmentally hazardous even at low concentrations, due to their properties of high toxicity, bioaccumulation, and non-degradability and their ubiquitous presence in nature [7]. Entering a water system, they can accumulate and be biomagnified to a degree in water, sediment, and the aquatic food chain, which may have toxic effects on aquatic biota [6,8]. Many metals form stable complexes with biomolecules, and this makes them potentially harmful to plants and animals even at low quantities [9]. HMs can also be transferred and accumulated in human tissues through long-term food consumption and water ingestion, causing hepato-, nephron-, neuro-, and genotoxic effects [6]. HMs are adsorbed and accumulated on the riverbed due to their low solubility in water and high particle affinity [5]. Bottom sediments provide habitats and a food source for benthic fauna [8], which may accumulate HMs from sediments. HMs can be taken up into fish from water, sediments, and food due to their position in the food chain. In order to maintain the quality of waters and avoid their contamination, continuous efforts have been made to develop technologies that are easy to use, sustainable, and cost-effective. Among different remediation techniques, phytoremediation has been proven to have the most effective approach to alleviate the environmental problems associated with contamination [10,11]. The present study is aimed to assess HM contamination, and its environmental effects and mitigation tools for an industrially affected river catchment basin.

To investigate the HM contamination in an industrially affected river catchment basin, the authors measured metal concentrations in the Armenian Debed River and some of its tributaries. Mining is one of few thriving economic sectors in Armenia [12]. Mining activities are highly concentrated in the Debed catchment basin. Generally, inefficient management and the ecologically unsafe practice of the removal of industrial wastewaters are among the most important environmental issues in the river basin and cause HM contamination in the rivers [13,14,15]. Monitoring HMs in the river waters and biota of the Debed catchment basin and conducting HM remediation experiments are, therefore, important steps toward the characterization of HM contamination, its environmental effects, remediation efficiency for an industrially affected river catchment basin, and overall regional sustainability.

## 2. Materials and Methods

### 2.1. Study Area

The Debed River catchment basin is located in northern Armenia (Lori Province, see Figure 1) and lies between 41°15′ and 40°41′ north latitude and 43°56′ and 44°57′ east longitude. It is presented by a combination of fold-fragment mountains, volcanic masses, highland plains and plateaus, intermountain concavities, and narrow and deep river valleys [16]. The surface area is 4080 km^2^. The rivers in northern Armenia belong to the Kura River basin. The Debed is a transboundary river and has a total length of 178 km, 152 km of which is in Armenia with the rest in the territory of Georgia [17].

### 2.2. River Field Sampling

Sampling was done in 8 locations of the rivers in the Debed catchment basin as outlined in Figure 1 and Table 1. Sampling sites were categorized into two groups: River sites P-1, D-2, and K-3 are located in the areas that do not have noticeable anthropogenic influences indicating the background state of the river basin; and river sites L-4, D-5, A-6, Sh-7, and D-8 are at risk of mining impact.

Water samples for the analysis of HMs were taken with polythene bottles pre-washed with 20% nitric acid (HNO_3_), as well as distilled water, in April, July, and September 2017 and preserved with concentrated HNO_3_. Overall, 24 water samples were collected from the study area under consideration.

Macrozoobenthos samples for HM analysis were gathered with a kick-net from different points in each investigated location in April 2017 and mixed to make one sample for each location. Benthic macroinvertebrates for the quantitative and qualitative analyses of animals were collected with a Surber sampler from 5 points in each location in April, July, and September 2017 and mixed to make one sample for each location. The animal samples for HM analysis were stored in cool boxes under low-temperature conditions, while the samples for biological investigation were preserved with a formaldehyde solution. In total, 8 mixed samples were taken for HM analysis and 24 mixed samples for biological analysis.

Fish samples (*Capoeta Capoeta* Guldenstadt 1773) for HM analysis were collected in April 2017 using a backpack electrofisher and stored in cool boxes under low-temperature conditions until the laboratory treatment of the samples. Fish were obtained only from 6 locations, as no one fish was registered in river sites L-4 and A-5. In addition, 3–5 samples of Kura scraper (*Capoeta Capoeta* Guldenstadt 1773) were gathered from each location. Overall, 25 fish samples were obtained for HM analysis. Kura scraper was selected for HM analysis because it is one of the widely distributed edible fish species in the Debed basin [18] and one of the most consumed species by local fishermen.

### 2.3. Sample Preparation and Analyses for HMs and Macrozoobenthos

In the laboratory, the water samples for HM analysis were digested on a hot plate using conc. HNO_3_ and H_2_O_2_ in a ratio of 4:1. The fish samples for HM analysis were frozen at about −20 °C until dissection, and the macrozoobenthos samples for HM analysis were washed with distilled water and separated from the substrate. The fish samples were dissected into gills, liver, and muscles, and the tissue samples obtained from scrapers collected from each location were mixed to make one sample for each organ. In total, 6 mixed samples from the 6 locations were obtained for each organ. The fish gills, liver, and muscles and benthic animals were then dried in an incubator at 50 °C. The biological samples were ground into powder by a mortar and pestle and digested on a hot plate using conc. HNO_3_ and H_2_O_2_ in a ratio of 3:2. The digested water and biological samples were analyzed for some HMs using an atomic absorption spectrometer (NovAA^®^ 350, Analytik Jena, Jena, Germany) for Cu, Mo, Cd, Pb (water and biological samples), Mn, and Zn (biological samples) [19], and a multi-parameter photometer (HI83200, Hanna Instruments, Woonsocket, RI, USA) for Mn, Zn (water samples), and Fe (water and biological samples) [20]. The water samples in many cases before measurement were evaporated to increase the density of HMs in the water, which made it possible to increase the sensitivity of HM measurements. Due to the limited measurement range of a photometer, the water samples in some cases were diluted to decrease the density of HMs in the water. All chemicals used were of analytical grade. Deionized water was used for the preparation of calibration standards and in the analyses. All glassware used were pre-washed with 10% HNO_3_, followed by rinsing with distilled water prior to use. To ensure that HM analyzers remained calibrated during the experiments, certified reference materials and certified standard solutions were analyzed for water and biological samples.

The fixed macrozoobenthos samples for quantitative and qualitative analyses were separated from the substrate and subsequently identified microscopically to taxonomic levels. The animals of each taxonomic group were dried on a filter paper and then quantified and weighed to obtain the total quantity and dry mass of each taxon.

### 2.4. Assessment of HM Contamination, Environmental Hazards, and Macrozoobenthos Diversity

The HMs content in the river waters and animals (macrozoobenthos and fish) was assessed with the geo-accumulation index (I_geo_) [21].
(1)$$I_{\mathrm{geo}}\;=\;\log_{2}\left(\frac{C_{m}}{1.5C_{b}}\right),$$
where C_m_ is the measured concentration of metal in water and biological samples, and C_b_ is the geochemical metal background value. The lowest registered value of each element was considered as background. The contamination degree based on I_geo_ values was classified according to [21] in Table 2.

Noncarcinogenic health risks associated with HMs in the river waters and fish were assessed with the risk assessment methodology adopted from the U.S. Department of Energy (USDOE, 2011) [22] and the U.S. Environmental Protection Agency (USEPA, 2011) [23]. The exposure doses through ingestion and dermal absorption were calculated using Equations (2)–(4):(2)$${\mathrm{ED}}_{w-\mathrm{ing}}=\frac{C_{w}\times {\mathrm{IngR}}_{w}\times \mathrm{ED}\times \mathrm{EF}}{\mathrm{BW}\times \mathrm{AT}},$$
(3)$${\mathrm{ED}}_{w-\mathrm{derm}}=\frac{C_{w}\times K_{p}\times \mathrm{ET}\times \mathrm{ED}\times \mathrm{EF}\times \mathrm{SA}\times {\mathrm{CF}}_{w}}{\mathrm{BW}\times \mathrm{AT}},$$
(4)$${\mathrm{ED}}_{f}=\frac{C_{f}\times \mathrm{ED}\times \mathrm{EF}\times {\mathrm{IngR}}_{f}\times {\mathrm{CF}}_{f}}{\mathrm{BW}\times \mathrm{AT}},$$
where ED_w-ing_ is the exposure dose through the ingestion of water (mg kg^−1^ day^−1^), ED_w-derm_ is the exposure dose through the dermal absorption of water (mg kg^−1^ day^−1^), ED_f_ is the exposure dose through the ingestion of fish (mg kg^−1^ day^−1^), C_w_ is the measured HM concentrations in water (mg L^−1^), C_f_ is the measured HM concentrations in fish (mg kg^−1^), IngR_w_ is the water ingestion rate for receptors (L day^−1^), IngR_f_ is the fish ingestion rate for receptors (kg meal^−1^), CF_w_ is the volumetric conversion factor (L cm^−3^), CF_f_ is the factor for the conversion of fresh fish to dry constant weight (unitless), ED is the exposure duration (year), EF is the exposure frequency (days year^−1^), K_p_ is the dermal permeability coefficient (cm h^−1^), ET is the exposure time (h day^−1^), SA is the skin surface area available for exposure (cm^2^), BW is the average body weight (kg), and AT is the averaging time for noncarcinogens (days). The average ingestion rate of fresh fish for consumers in the Debed basin was assessed based on our questionnaire of fishermen, the families of which are the main consumers of the river fish.

The noncarcinogenic hazard quotient was calculated by Equations (5)–(7):(5)$${\mathrm{HQ}}_{w-\mathrm{ing}}=\frac{{\mathrm{ED}}_{w-\mathrm{ing}}}{{\mathrm{RfD}}_{w-\mathrm{ing}}},$$
(6)$${\mathrm{HQ}}_{w-\mathrm{derm}}=\frac{{\mathrm{ED}}_{w-\mathrm{derm}}}{{\mathrm{RfD}}_{w-\mathrm{derm}}},$$
(7)$${\mathrm{HQ}}_{f}=\frac{{\mathrm{ED}}_{f}}{{\mathrm{RfD}}_{f}},$$
where HQ_w-ing_ is the hazard quotient via ingestion of water (unit less), HQ_w-derm_ is the hazard quotient via dermal contact with water (unitless), HQ_f_ is the hazard quotient via ingestion of fish (unitless), RfD_w_ is the ingestion reference dose for water (mg kg^−1^ day^−1^), RfD_w-derm_ is the dermal reference dose for water (mg kg^−1^ day^−1^), and RfD_f_ is the ingestion reference dose for fish (mg kg^−1^ day^−1^). RfD values were derived from USEPA (2003) [24].

Overall noncarcinogenic effects posed by all metals, expressed as the hazard index (HI) via ingestion contact with water/fish and dermal contact with water, were assessed by the following Equation:(8)$${\mathrm{HI}}_{w-\mathrm{ing}/w-\mathrm{derm}/f}=\sum_{i=0}^{n}{\mathrm{HQ}}_{w-\mathrm{ing}/w-\mathrm{derm}/f}.$$

Diversity of the macrozoobenthos community in the river ecosystems was assessed with Margalef’s richness index (MRI) [25].
(9)$$\mathrm{MRI}=\frac{\left(S-1\right)}{\mathrm{lnN}},$$
where S is the total number of species in a sample and N is the total number of individuals in the sample. Statistical analyses of the results obtained from this study were done using Statistica 8 and Microsoft Excel 2019 software programs.

### 2.5. Phytoextraction of HMs

For the preparation of the experimental environment, the individual stock solutions of HMs, which were made by adding an appropriate amount of individual metal salts to ultrapure water, were mixed with Waris-H medium. A scientific experiment was carried out in the experimental solution (Table 3) and its 1:10 and 1:100 dilutions, as well as a control medium (without adding extra quantities of HMs to Waris-H medium). The latter was necessary for clarifying the vitality and growth intensity of different strains. Then, the pH of media was measured with a multiparameter instrument (Multi Lab P5, WTW, Weilheim, Germany), adjusted to 7.0 with 1 M NaOH, and the media were sterilized in an autoclave. The experimental strains *Desmodesmus abundans* M3456, *Stichococcus bacillaris* M1898, and *Calothrix desertica* M0455 were provided by the Culture Collection of Algae at the University of Cologne, Germany (www.ccac.uni-koeln.de (accessed on 29 January 2021). The chosen strains had been isolated from waste drainage, which contains high amounts of HMs; thus, the strains have a relatively high tolerance to HMs in order to survive in such an environment. The microalgae cultures were grown in Waris-H culture medium at 23 °C and fluorescent white light of 20–40 µmol m^−2^ s^−1^ using a light/dark cycle of 14/10 for 24 days. The experimental media were aerated with a mixture of filtered air and CO_2_ (about 1% *v*/*v*). For the revelation of the efficiency of HM removal by microalgae and of microalgae species with the highest capacities of bioaccumulation of HMs, the experimental media with suspension cultures were centrifuged, and 1 mL supernatant was taken and analyzed for HMs (Mn, Pb, Cu, Zn, Cd, Mo) with inductively coupled plasma mass spectrometry by the Lipidomics/Metabolomics Facility of University of Cologne, Germany (https://www.cecad.uni-koeln.de/research/core-facilities/lipidomicsmetabolomics-facility (accessed on 29 January 2021). For the revelation of HM-resistant microalgae strains, dry weight measurements were implemented. pH measurements were also performed. The biomass was determined by filtering the experimental media through pre-dried and pre-weighed membrane filters (Whatman ME 28, Little Chalfont, UK) and weighing the filters after being dried in an oven at 105 °C for one hour, and the pH was determined with a multiparameter instrument (Multi Lab P5, WTW, Weilheim, Germany). The experiments were made in triplicate, and the data, as obtained, were averaged.

The tolerance of *D. abundans* M3456 against HMs was determined by growing the mentioned strain in a solution with maximum possible concentrations of the HMs. The solution was prepared by maintaining the ratio of the metals present in the medium, but the concentrations were increased so that the metals did not precipitate. As a result, compared to the experimental medium, a 5 times denser solution was prepared. The strain was also parallelly grown in the control medium. The strain was grown according to the above-described procedure except the duration, which lasted 7 days. The number of cells was counted in a Neubauer counting chamber. The experiments were performed in triplicate, and the data, as obtained, were averaged.

## 3. Results and Discussion

### 3.1. HMs in River Water

The spatial distribution patterns of HM concentrations in the river waters of the Debed catchment basin are shown in Figure 2. The average Fe concentration in water (Figure 2) was significantly higher (*p* ˂ 0.001) in sites D-5, A-6, and D-8 compared to the other investigated sites. The highest average value of Fe was registered in Debed River site D-5 located downstream of Alaverdi Town, approximately 2.3 times higher compared to background site D-2, located upstream of Alaverdi Town (Figure 2). This indicates that the Debed River water in site D-5 was contaminated with Fe in the territory of Alaverdi Town where the potential contamination source of Fe involves the discharges from the Alaverdi copper smelter activity. Other smaller sources of Fe contamination in the Debed basin can be the discharges from the Akhtala mountain enrichment combine (including tailing dump) activity, which caused a significantly higher (*p* ˂ 0.001) average Fe concentration in Akhtala River site A-6 located downstream of the combine, compared to the background sites (P-1, D-2, and K-3). The aforementioned anthropogenic factors also caused Fe contamination in Debed River site D-8 located downstream of Debed River site D-5 and the confluence of the Akhtala and Debed rivers (Figure 2).

The average Cu concentration in water (Figure 2) was significantly higher (*p* ˂ 0.001) in river sites L-4, A-6, and Sh-7 compared to the background sites. The highest average Cu concentration was recorded in site A-6, approximately 15–23 times higher compared to the background sites (Figure 2).

All of this indicates that the Akhtala mountain enrichment combine activity also caused Cu contamination in Akhtala River site A-6, while the water in Lalvar River site L-4 located downstream of the nonoperational mining areas of Alaverdi was contaminated with Cu (Figure 2), supposedly caused by the effluents from the nonoperational tailing dump of the former Alaverdi mining and metallurgical combine, as well as the landfills of the nonoperational Alaverdi underground copper mine. In addition, the water in Shnogh River site Sh-7 located downstream of the Teghut mountain enrichment combine was contaminated with Cu (Figure 2), supposedly originated from the Teghut mountain enrichment combine (including tailing dump) activity.

The average Zn concentration in water (Figure 2) was significantly higher (*p* ˂ 0.001) in river sites L-4, D-5, A-6, and D-8 compared to the other sites. The highest average Zn concentration was registered in Akhtala River site A-6, approximately 48–111 times higher compared to the background sites (Figure 2), indicating Zn origination from the same potential anthropogenic source as Cu. However, river sites L-4, D-5, and D-8 also showed visible contamination with Zn (Figure 2). The Zn contamination in site L-4 can be induced by the same potential sources as Cu, while site D-5 can be contaminated with Zn originated from the Alaverdi copper smelter activity and site D-8 with the latter and the pressure of the Akhtala tributary.

The spatial distribution pattern of average Mn concentration was similar to Cu and Zn with the highest registered average concentration in site A-6 (Figure 2). This indicates that it originated from the same potential anthropogenic sources as Cu and Zn. The highest average Mo and Pb concentrations of 0.068 and 0.0096 mg L^−1^, accordingly, were registered in river site A-6, while either trace concentrations or the nonregistration of these metals was in the background sites (Figure 2). There were Mo and Pb contamination also in sites L-4, Sh-7, and D-8, as well as Pb contamination in site D-5 (Figure 2). The Mo and Pb contamination in these sites can be induced by the same potential anthropogenic sources for each site as Cu and Zn.

We observed Cd contamination only in site A-6 potentially affected by the Akhtala mountain enrichment combine activity, while the other investigated sites showed either trace concentrations or nonregistration of this metal (Figure 2). The high standard deviation (SD) for the average concentrations of most of the investigated HMs in site A-6, where the highest degrees of contamination with almost all the investigated elements were registered (Figure 2), indicates a wide range of the concentrations of these metals, which is explained by uncontrolled point and nonpoint contamination potentially induced by the Akhtala mountain enrichment combine activity.

Pearson’s correlation analysis showed a strong correlation (r > 0.7) among the average concentrations of Zn, Mn, Cu, Pb, and Cd according to the river observation sites (Table 4), indicating that these metals likely originated from the same anthropogenic sources in each investigated site. Fe and Mo did not show such a correlation with the other metals (Table 4), which was probably due to the fact that not all the anthropogenic sources of other metals caused Fe and Mo contamination.

### 3.2. HMs in River Macrozoobenthos and Fish

The results of the analysis of HMs in the tissues of macrozoobenthos and fish (Kura scrapers) collected from the rivers in the Debed catchment basin showed a noticeable increase in the concentrations of different HMs in river sites L-4, D-5, A-6, Sh-7, and D-8 being under mining impact, compared to the background sites (Table 5). 

Data on HMs in fish were not obtained for sites L-4 and A-6, because fishes were not registered at these sites. The Fe concentration in macrozoobenthos tissues (Table 5) was significantly higher (*p* ˂ 0.001) in sites D-5, A-6, and D-8 compared to the other investigated sites. This distribution pattern of Fe concentration in the tissues of benthic animals (Table 5) has similarity with the distribution pattern of average concentration of this metal in the river waters (Figure 2). The Cu concentration in macrozoobenthos tissues (Table 5) was significantly higher (*p* ˂ 0.01) in sites L-4, D-5, and A-6, compared to the other investigated sites. High concentrations of Zn were registered in the tissues of macrozoobenthos from site A-6 and the liver of fish from site D-8, approximately 7–18 (macrozoobenthos) and 3.3–3.5 (fish) times higher compared to the background sites (Table 5). The Mn concentration in macrozoobenthos tissues (Table 5) was significantly higher (*p* ˂ 0.01) in sites L-4, D-5, and A-6 compared to the other investigated sites. The Mo concentration in macrozoobenthos tissues was only registered in sites D-5 and Sh-7; however, the concentration in site Sh-7 was approximately 4 times higher compared to the other site (Table 5). The Mo concentration in the fish organs was only registered in sites D-5, Sh-7, and D-8 (Table 5). The visible concentration of Pb in macroinvertebrate tissues was only registered in sites L-4, D-5, and A-6, while the animals in the other sites were characterized by either trace concentrations or nonregistration of this metal (Table 5). Overall, high concentrations of different HMs among Fe, Cu, Zn, Mn, Mo, and Pb in the hydrobionts were observed in river sites L-4, D-5, A-6, Sh-7, and D-8 (Table 5) where the waters were also characterized by high levels of these metals (Figure 2), which indicates the approximate accumulation of these elements from the waters into the aquatic animals.

### 3.3. Assessment of HM Contamination in River Waters, Macrozoobenthos, and Fish

Assessment of HM contents in the water and biological samples was done according to Igeo based on the lowest registered value as background. The water contamination degrees were assessed based on the average values of HMs. The results of the assessment are given in Table 6 and Table 7. Pollution was considered as those cases that exceeded class 1 of contamination. According to the results (Table 6), the waters only in river sites L-4, D-5, A-6, Sh-7, and D-8 being under mining impact were polluted with different investigated HMs, while the macrozoobenthos in these sites was noticeably more polluted with different metals, compared to the background sites. The rivers also showed fish (Kura scraper) pollution in mining impact sites D-5, Sh-7, and D-8. The fish in site D-5 were polluted with Fe (gills) and Mn (liver), the fish in site Sh-7 with Mn (liver), and the fish in site D-8 with Zn (liver) (Table 7). The degrees of this contamination in the waters, macrozoobenthos, and fish of the aforementioned mining impact sites ranged between class 2 (moderately contaminated) and class 6 (extremely contaminated), while the background sites rarely showed pollution evaluated as class 2 in almost all the registered cases (Table 6 and Table 7). The investigated river sites can be ranked according to a contamination degree decreasing order as follows: A-6–D-5–L-4–D-8–Sh-7–P-1/D-2/K-3 for water, A-6–D-5–L-4–Sh-7–D-8–D-2/K-3–P-1 for macrozoobenthos (Table 6), and K-3–D-2–D-5–Sh-7–D-8–P-1 for fish (Table 7).

### 3.4. Adverse Health Effects by HM Contamination in River Water and Fish

The results of the calculation of HI through the ingestion of river waters and fish and the dermal absorption of river waters are shown in Figure 3.

The results show that the HI through dermal contact with river waters and the ingestion of river fish (Kura scrapers) was lower than the threshold value of 1 (Figure 3), which indicates that the adverse effects of the investigated HMs on human health in the case of waters used for domestic (bathing/showering) purposes and ingestion of fish were negligible. Meanwhile, the average value of HI through the ingestion of river water by children was greater than 1, particularly in site A-6 (Figure 3). Therefore, the investigated HMs in this site are considered to have the probability of adverse health effects on children in the case of river water use for drinking purposes. However, the SD for the average values of HI indicates that the concentrations of the investigated HMs in sites A-6 and Sh-7 in some months were also risky for adults’ and children’s health, respectively, in the case of waters used for drinking purposes (Figure 3). According to the average values of the hazard quotient (HQ), the health hazards of individual HMs can be ranked as Mn > Cd > Cu > Pb > Mo > Fe > Zn and Mo > Mn > Cu > Fe > Pb > Zn > Cd in sites A-6 and Sh-7, respectively. The SD for the average values of HQ indicates that the concentrations of even single elements such as Mn (average HQ value of 0.98 and SD of 0.15) and Cd (average HQ value of 0.36 and SD of 0.62) in site A-6 in some months were risky for children’s health in the case of water used for drinking purposes.

### 3.5. Adverse Biological Effects by HM Contamination in River

The average values of macrozoobenthos quantitative and qualitative parameters were noticeably lower in the mining impact sites (L-4, D-5, A-6, Sh-7, and D-8) compared to the background sites (P-1, D-2, and K-3), as shown in Figure 4.

The lowest average macrozoobenthos abundance of 34 ind. m^−2^ was registered in site A-6, approximately 10–133 times lower compared to the background sites (Figure 4). The lowest average biomass of 0.056 mg m^−2^ was observed in site L-4, approximately 46–273 times lower compared to the background sites (Figure 4). An average species number of 3 from the orders Ephemeroptera, Diptera, and Lumbricina was registered in site L-4; average species number of 5 from the orders Haplotaxida, Plecoptera, Diptera, and Trichoptera in site D-5; average species number of 3 from the orders Diptera, Hemiptera, and Lumbricina in site A-6; average species number of 5 from the orders Ephemeroptera, Diptera, Plecoptera, and Trichoptera in site Sh-7; and average species number of 6 from the orders Haplotaxida, Odonata, Ephemeroptera, Diptera, Heteroptera, and Trichoptera in site D-8, while the background sites were characterized by a much richer qualitative composition, with average species numbers between 11 and 18 (Figure 4). The river macrozoobenthos diversity status, expressed as MRI, is presented in Figure 4. The investigated river sites can be ranked according to an MRI decreasing order as follows: K-3–D-2–P-1–D-8–Sh-7–D-5–L-4/A-6. The macrozoobenthos diversity was most affected in sites L-4 and A-6, where the average values of MRI were approximately 3–5 times lower compared to the background sites (Figure 4).

All the macrozoobenthos parameters showed a moderate negative correlation (0.3 ˂ *r* ˂ 0.7) with almost all the investigated HMs in the river waters according to the observation sites (Table 8). A similar pattern of correlation was also observed between macrozoobenthos parameters and HMs in macrozoobenthos (Table 9). All of this indicates that macrozoobenthos growth was adversely affected by HM contamination in the rivers of the Debed catchment basin.

### 3.6. Phytoextraction of HMs in Water

From the strains chosen, the highest biomass attained was observed for *D. abundans* M3456 under all experimental treatments (Figure 5). By consequence, the pH at the end of the experiment was also highest in the *D. abundans* culture (average pH value of about 11 compared to average pH value of about 8 in the other two algae cultures, as shown in Figure 5).

The HM analysis has revealed that there is a high-capacity uptake of HMs, particularly Mn, Pb, Cu, Zn, and Cd, from the culture media by different strains. There were differences observed in HM uptake both among the algae and among the metals, shown in Figure 6.

The highest value of uptake of HMs was registered for the strain *D. abundans* M3456 and for the metals Mn and Pb (100%; Figure 6). Biosorption of HMs with *D. abundans* M3456 occurred in the following order: Mn = Pb > Cu > Zn = Cd > Mo (Figure 6). To determine the tolerance of *D. abundans* M3456 against these HMs, the mentioned strain was grown in a solution with maximum possible concentrations of the HMs and the control medium. A noticeable difference between strain growths in the control and experimental media was not observed (Figure 7), which indicates that *D. abundans* M3456 is quite tolerant of HMs.

The investigated rivers were polluted with these metals (Table 6 and Table 7) induced by mining activities, which indicates that the phytoextraction technique with the use of *D. abundans* M3456 can be efficiently used for the removal of some emerging contaminants from mining wastewaters discharged in the Debed catchment basin. This is a valuable finding for the planning of the phytoremediation technique for the treatment of mining wastewaters in the Debed basin.

## 4. Conclusions and Recommendations

The present study provides valuable information about HM contamination in a river catchment basin suffering from an abundance of mining activities, and its hydro-ecological and associated health effects. The evaluation of HM contamination in water, macrozoobenthos, and fish showed that the rivers, in one of the most developed Armenian mining areas, the Debed catchment basin, were contaminated with several HMs at levels that caused macrozoobenthos diversity and biomass decline, and human health risks in the case of waters used for drinking purposes; however, there were no health risks for fish (Kura scraper) consumption by local people in the conditions of evaluated fish ingestion rate. The study has indicated that the microalgal remediation of mining wastewaters, especially with *D. abundans* M3456, is an efficient tool to mitigate river contamination with such HMs as Mn, Pb, Cu, Zn, and Cd that were at alarming levels in the rivers of the Debed catchment basin. The results from this study can serve as insight for environmental managers in contamination mitigation practices and river basin management for the future.

In order to prevent or mitigate the HM contamination in the Debed River catchment basin, it is recommended to implement the following measures:Setting of new treatment technologies (e.g., microalgal remediation) for improving the quality of wastewaters discharged from the mining industry.Implementation of remediation works (i.e., covering with soil and vegetation) in the areas affected by the mining industry.Maintenance of the rules for technical operation of the tailing dumps and restoration of their damaged parts.Definition of strict regulations for waste storage and emissions for miners.

## Figures and Tables

**Figure 1 ijerph-18-02881-f001:**
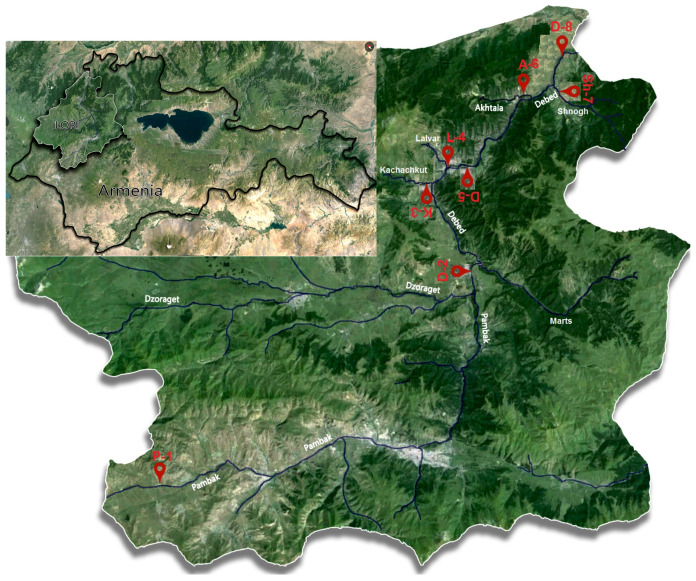
Location of Debed River catchment basin in Armenia and the river monitoring sites as coded in Table 1.

**Figure 2 ijerph-18-02881-f002:**
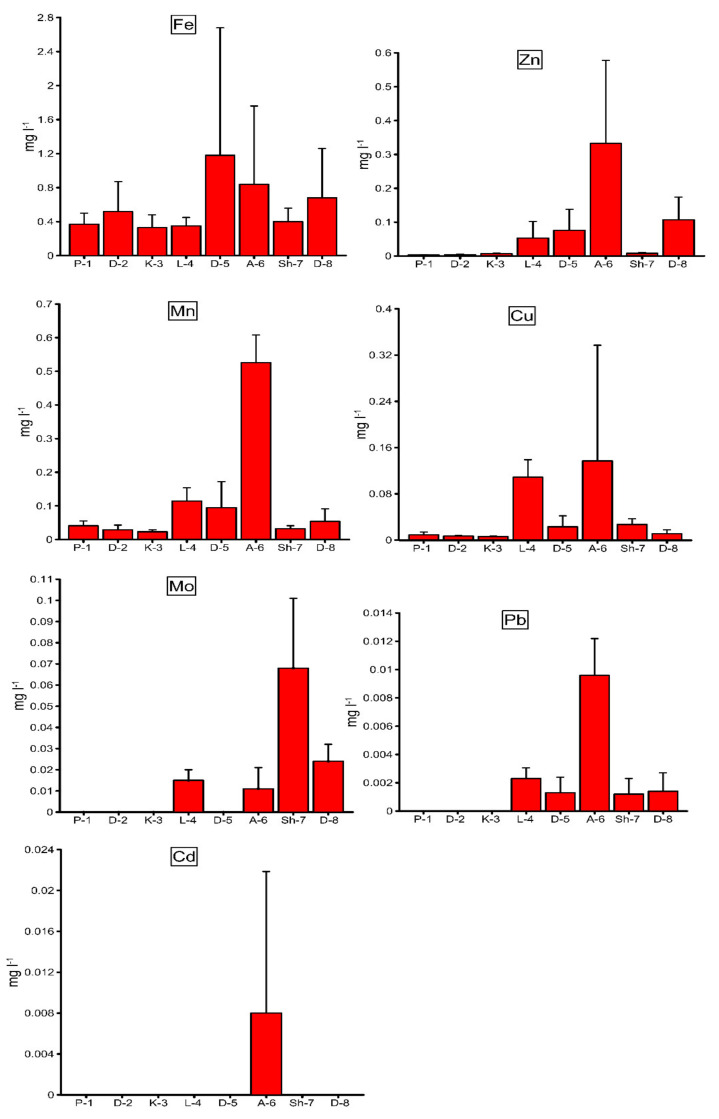
Spatial distribution of the average concentrations of some HMs in the river waters of the Debed catchment basin.

**Figure 3 ijerph-18-02881-f003:**
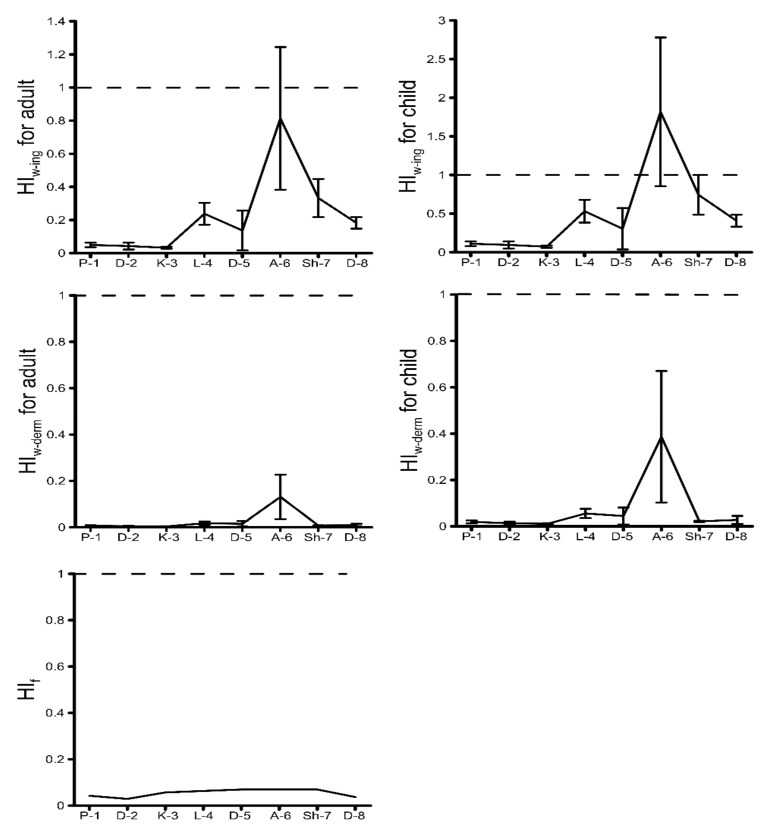
Hazard index (HI) of HMs in the river waters and fish of the Debed catchment basin.

**Figure 4 ijerph-18-02881-f004:**
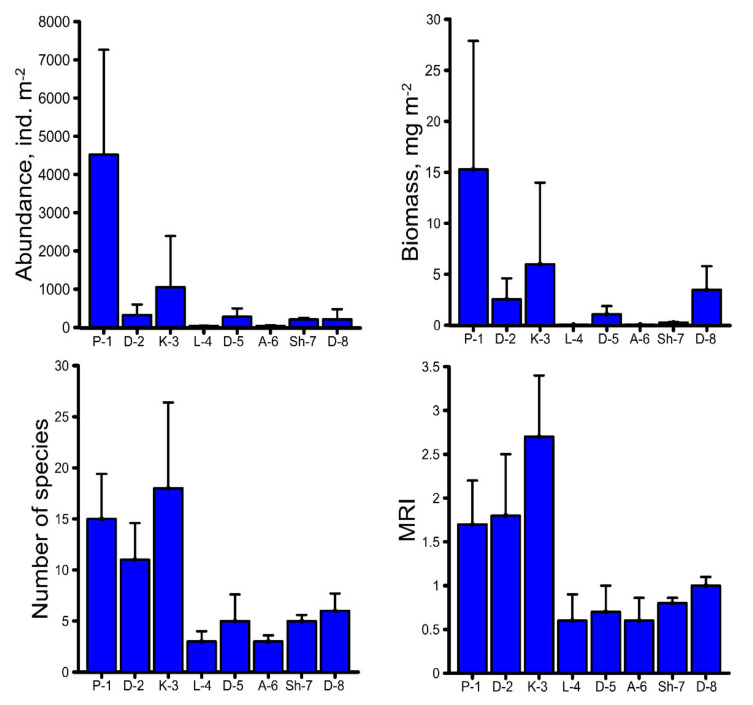
Spatial distribution of the average values of macrozoobenthos parameters in the rivers of the Debed catchment basin.

**Figure 5 ijerph-18-02881-f005:**
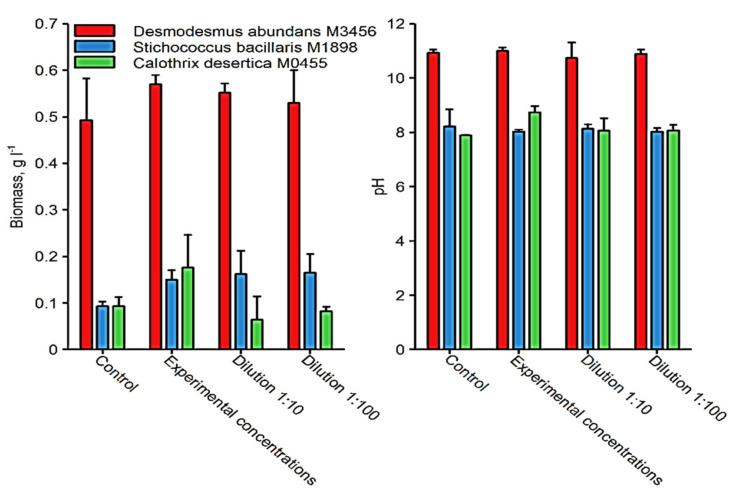
Average values of biomass and pH of different strains in diverse treatment conditions.

**Figure 6 ijerph-18-02881-f006:**
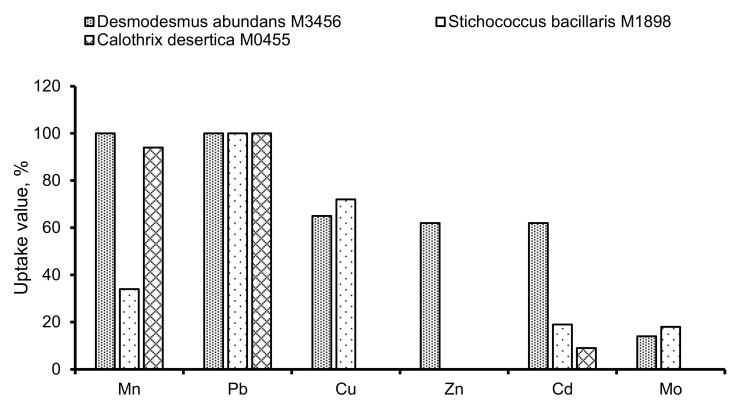
Values of uptake of HMs from supernatant for diverse strains.

**Figure 7 ijerph-18-02881-f007:**
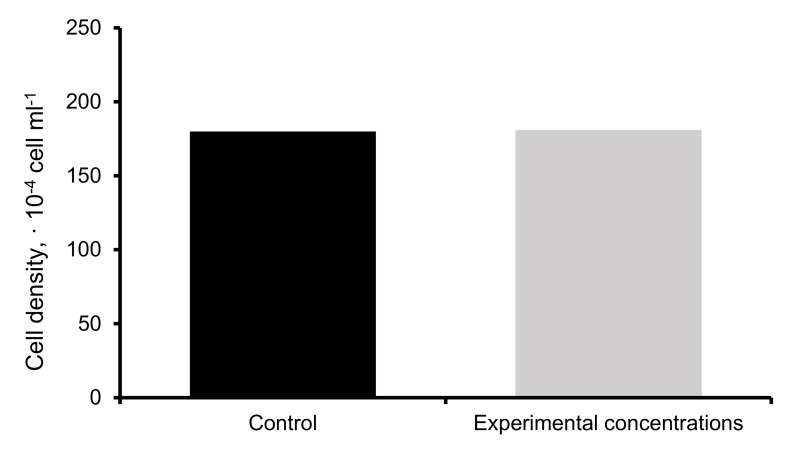
Average values of cell density of *D. abundans* M3456 in different treatment conditions.

**Table 1 ijerph-18-02881-t001:** Coordinates of sampling sites in the Debed River catchment basin.

Sampling Site Code	N/Lat	E/Long	River Site Location
P-1	40°50′51.8″	44°09′36.6″	Pambak River site located in the river upstream
D-2	40°59′08.2″	44°39′04.8″	Debed River site located downstream of the confluence of the Pambak and Dzoraget rivers
K-3	41°04′55.2″	44°37′07.2″	Kachachkut River site located near the river mouth
L-4	41°05′56.8″	44°39′28.8″	Lalvar River site located near the river mouth
D-5	41°06′48.5″	44°44′16.2″	Debed River site located downstream of Alaverdi Town
A-6	41°08′57.3″	44°46′56.0″	Akhtala River site located in the river downstream
Sh-7	41°09′04.1″	44°49′49.3″	Shnogh River site located near the river mouth
D-8	41°12′33.4″	44°54′08.8″	Debed River site located near Ayrum Town

**Table 2 ijerph-18-02881-t002:** Classification of contamination degree based on I_geo_ values.

I_geo_ Value	Contamination Degree	Contamination Class
˂0	Uncontaminated	0
≥0, ˂1	Uncontaminated to moderately contaminated	1
≥1, ˂2	Moderately contaminated	2
≥2, ˂3	Moderately to heavily contaminated	3
≥3, ˂4	Heavily contaminated	4
≥4, ˂5	Heavily to extremely contaminated	5
≥5	Extremely contaminated	6

**Table 3 ijerph-18-02881-t003:** Heavy metal (HM) compounds and their experimental concentrations.

Metal Compound	MnCl_2_·2H_2_O	NiCl_2_·6H_2_O	Cr_2_K_2_O_7_	Na_2_MoO_4_·2H_2_O	VCl_3_	ZnCl_2_	CuSO_4_ ·5H_2_O	PbCl_2_	CdCl_2_	Co(NO_3_)_2_·6H_2_O	NaAsO_2_
Concentration in environment (µg L^−1^)	191.7	2.8	2.3	508.4	2.9	108.0	120	0.9	0.8	1.8	2.03

**Table 4 ijerph-18-02881-t004:** Pearson’s coefficients of correlations between HMs.

HMs	Fe	Zn	Mn	Cu	Mo	Pb	Cd
Fe	1						
Zn	0.53	1					
Mn	0.41	0.96	1				
Cu	0.14	0.75	0.84	1			
Mo	−0.22	−0.06	−0.09	0.04	1		
Pb	0.39	0.97	0.99	0.86	0.04	1	
Cd	0.34	0.94	0.98	0.75	−0.06	0.97	1

**Table 5 ijerph-18-02881-t005:** Spatial distribution of HM concentrations (mg kg^−1^) in the river macrozoobenthos and fish of the Debed catchment basin.

Sampling Sites	Fe	Zn	Mn	Cu	Mo	Pb	Cd
Macrozoobenthos tissues							
P-1	204.75	71.50	66.30	21.13	NR	NR	NR
D-2	1005.00	195.00	256.00	39.50	NR	Traces	NR
K-3	750.29	200.57	110.31	74.29	NR	NR	NR
L-4	517.36	409.57	274.22	175.85	NR	1.40	NR
D-5	6364.18	329.85	395.82	203.73	5.82	1.36	NR
A-6	5066.67	1311.11	284.44	133.33	NR	1.93	NR
Sh-7	627.14	272.33	105.37	104.27	21.89	Traces	NR
D-8	1827.03	165.14	107.86	86.43	NR	Traces	NR
Kura scraper gills							
P-1	132.95	97.50	22.69	4.20	NR	NR	NR
D-2	243.92	115.26	47.98	6.86	NR	NR	NR
K-3	216.67	117.62	49.03	10.52	Traces	NR	NR
D-5	222.60	113.97	36.51	8.90	1.78	NR	NR
Sh-7	168.47	77.38	31.28	11.93	0.94	NR	NR
D-8	65.80	44.94	45.10	6.58	NR	NR	NR
Kura scraper liver							
P-1	431.05	95.79	17.11	23.4	NR	NR	NR
D-2	469.13	96.09	40.53	20.91	NR	NR	NR
K-3	364.00	104.00	53.17	33.80	NR	NR	NR
D-5	704.74	95.79	35.03	25.32	NR	NR	NR
Sh-7	421.76	86.27	25.34	38.41	5.71	NR	NR
D-8	239.32	339.77	7.39	27.77	14.77	NR	NR
Kura scraper muscles							
P-1	90.43	50.87	4.71	4.01	NR	NR	NR
D-2	90.10	47.62	11.84	3.22	NR	NR	NR
K-3	116.07	27.57	11.14	3.05	NR	NR	NR
D-5	118.30	49.40	6.66	4.16	1.30	NR	NR
Sh-7	98.43	32.84	6.34	4.84	1.70	NR	NR
D-8	144.16	30.89	3.22	3.06	NR	NR	NR

“NR”—nonregistration.

**Table 6 ijerph-18-02881-t006:** HM contamination degrees in the river waters and macrozoobenthos of the Debed catchment basin.

Sampling Sites	Indicator Metals	Contamination Degree (Class)	Indicator Metals	Contamination Degree (Class)
	Water		Macrozoobenthos	
P-1	Fe, Zn, Cu, Mo, Pb, Cd	UC (0)	Fe, Zn, Mn, Cu, Mo, Pb, Cd	UC (0)
	Mn	UC-MC (1)		
D-2	Zn, Mn, Cu, Mo, Pb, Cd	UC (0)	Mo, Cd	UC (0)
			Pb	≥UC (≥0)
			Zn, Cu	UC-MC (1)
	Fe	UC-MC (1)	Fe, Mn	MC (2)
K-3	Fe, Mn, Cu, Mo, Pb, Cd	UC (0)	Mo, Pb, Cd	UC (0)
			Zn, Mn	UC-MC (1)
	Zn	UC-MC (1)	Fe, Cu	MC (2)
L-4	Fe	UC (0)	Mo, Cd	UC (0)
	Mo, Cd	≥UC (≥0)	Fe	UC-MC (1)
	Pb	≥UC-MC (≥1)	Zn, Mn	MC (2)
	Mn	MC (2)	Pb	≥MC (≥2)
	Zn, Cu	HC (4)	Cu	MC-HC (3)
D-5	Mo, Cd	UC (0)	Cd	UC (0)
			Mo	≥UC-MC (≥1)
	Pb	≥UC (≥0)	Zn, Mn	MC (2)
			Pb	≥MC (≥2)
	Fe, Mn, Cu	MC (2)	Cu	MC-HC (3)
	Zn	HC-EC (5)	Fe	HC-EC (5)
A-6	Fe	UC-MC (1)	Mo, Cd	UC (0)
	Mo, Pb, Cd	≥MC-HC (≥3)	Mn	MC (2)
	Mn, Cu	HC (4)	Pb	≥MC (≥2)
			Cu	MC-HC (3)
	Zn	EC (6)	Zn	HC (4)
			Fe	HC-EC (5)
Sh-7	Fe, Mn	UC (0)	Cd	UC (0)
	Pb, Cd	≥UC (≥0)	Pb	≥UC (≥0)
	Zn	UC-MC (1)	Mn	UC-MC (1)
	Mo	≥UC-MC (≥1)	Fe, Zn, Cu	MC (2)
	Cu	MC (2)	Mo	≥MC-HC (≥3)
D-8	Pb, Cd	≥UC (≥0)	Mo, Cd	UC (0)
			Pb	≥UC (≥0)
	Fe, Mn, Cu	UC-MC (1)	Zn, Mn	UC-MC (1)
	Mo	≥UC-MC (≥1)	Cu	MC (2)
	Zn	HC-EC (5)	Fe	MC-HC (3)

“UC”—uncontaminated; “UC-MC”—uncontaminated to moderately contaminated; “MC”—moderately contaminated; “MC-HC”—moderately to heavily contaminated; “HC”—heavily contaminated; “HC-EC”—heavily to extremely contaminated; “EC”—extremely contaminated.

**Table 7 ijerph-18-02881-t007:** HM contamination degrees in the river fish of the Debed catchment basin.

Sampling Sites	Indicator Metals	Contamination Degree (Class)	Indicator Metals	Contamination Degree (Class)	Indicator Metals	Contamination Degree (Class)
	Kura Scraper Gills		Kura Scraper Liver		Kura Scraper Muscles	
P-1	Mn, Cu, Mo, Pb, Cd	UC (0)	Zn, Cu, Mo, Pb, Cd	UC (0)	Fe, Mn, Cu, Mo, Pb, Cd	UC (0)
	Fe, Zn,	UC-MC (1)	Fe, Mn	UC-MC (1)	Zn	UC-MC (1)
D-2	Mo, Pb, Cd	UC (0)	Zn, Cu, Mo, Pb, Cd	UC (0)	Fe, Cu, Mo, Pb, Cd	UC (0)
			Fe	UC-MC (1)		
	Zn, Mn, Cu	UC-MC (1)	Mn	MC (2)	Zn	UC-MC (1)
	Fe	MC (2)			Mn	MC (2)
K-3	Pb, Cd	UC (0)	Zn, Mo, Pb, Cd	UC (0)	Fe, Zn, Cu, Mo, Pb, Cd	UC (0)
	Mo	≥UC (≥0)				
	Zn, Mn, Cu	UC-MC (1)	Fe, Cu	UC-MC (1)		
	Fe	MC (2)	Mn	MC-HC (3)	Mn	MC (2)
D-5	Pb, Cd	UC (0)	Zn, Cu, Mo, Pb, Cd	UC (0)	Fe, Cu, Pb, Cd	UC (0)
	Mo	≥UC (≥0)				
	Zn, Mn, Cu	UC-MC (1)	Fe	UC-MC (1)	Mo	≥UC (≥0)
	Fe	MC (2)	Mn	MC (2)	Zn, Mn	UC-MC (1)
Sh-7	Mn, Pb, Cd	UC (0)	Zn, Pb, Cd	UC (0)	Fe, Zn, Pb, Cd	UC (0)
	Mo	≥UC (≥0)	Mo	≥UC (≥0)		
			Fe, Cu	UC-MC (1)	Mo	≥UC (≥0)
	Fe, Zn, Cu	UC-MC (1)	Mn	MC (2)	Mn, Cu	UC-MC (1)
D-8	Fe, Zn, Mo, Pb, Cd	UC (0)	Fe, Mn, Cu, Pb, Cd	UC (0)	Zn, Mn, Cu, Mo, Pb, Cd	UC (0)
			Mo	≥UC-MC (≥1)		
	Mn, Cu	UC-MC (1)	Zn	MC (2)	Fe	UC-MC (1)

“UC”—uncontaminated; “UC-MC”—uncontaminated to moderately contaminated; “MC”—moderately contaminated; “MC-HC”—moderately to heavily contaminated.

**Table 8 ijerph-18-02881-t008:** Pearson’s coefficients of correlations between macrozoobenthos parameters and HMs in water.

Macrozoobenthos Parameters	Fe	Zn	Mn	Cu	Mo	Pb	Cd
Abundance	−0.35	−0.35	−0.26	−0.37	−0.37	−0.35	−0.21
Biomass	−0.37	−0.38	−0.34	−0.49	−0.46	−0.43	−0.28
Species number	−0.48	−0.54	−0.49	−0.65	−0.53	−0.58	−0.37
MRI	−0.49	−0.51	−0.46	−0.61	−0.53	−0.54	−0.34

**Table 9 ijerph-18-02881-t009:** Pearson’s coefficients of correlations between macrozoobenthos parameters (April) and HMs in macrozoobenthos.

Macrozoobenthos Parameters	Fe	Zn	Mn	Cu	Mo	Pb
Abundance	−0.39	−0.48	−0.56	−0.59	−0.28	−0.56
Biomass	−0.44	−0.46	−0.58	−0.62	−0.38	−0.62
Species number	−0.34	−0.47	−0.51	−0.54	−0.23	−0.58
MRI	−0.27	−0.36	−0.48	−0.56	−0.25	−0.53

## Data Availability

The data are contained within the text of this report.

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
