# Peer review of "Heavy Metal Contamination in an Industrially Affected River Catchment Basin: Assessment, Effects, and Mitigation"

_ijerph, 2021, doi:10.3390/ijerph18062881_

Round 1

Reviewer 1 Report

The manuscript ijerph-1120644 entitled “Heavy Metal Contamination in an Industrially Affected River Catchment Basin: Assessment, Effects and Mitigation” aimed to assess the concentration of Fe, Zn, Mn, Cu, Mo, Pb, Cd in waters, macrozoobenthos and fish from one of the most developed mining areas in Armenia. Furthermore, hydro-ecological and health effects were also assessed. The topic is of great interest, but the MS lacks novelty.

Generally, a lot of data are reported, even if it seems to be written at full speed with little overall processing of the ideas stated by the authors.

In particular, the main concern regards the M&M section. M&M should be described with sufficient detail to allow others to replicate and build on published results. The number of samples analysed was not declared. How many water, macroinvertebrates and fish samples did you collect?

The second concern regards the macroinvertebrates and fish samples. In literature it is well known that the trophic role influences metal accumulation. What kind of macroinvertebrates/fish species were selected? The terms “macrozoobenthos” or “fish” are too generic.

The citations and references are not well written. The authors didn’t follow the authors guidelines.

The third concern regards the lack of data from sediment (one of the main sink of metals in rivers).

I am sorry, and I know this is disappointing for the authors, but I have to say that this work is not suitable for a high-quality journal as IJERPH. Finally, the manuscript is not readable and must be linguistically checked (in particular a lot of prepositions are missing).

Author Response

The manuscript ijerph-1120644 entitled “Heavy Metal Contamination in an Industrially Affected River Catchment Basin: Assessment, Effects and Mitigation” aimed to assess the concentration of Fe, Zn, Mn, Cu, Mo, Pb, Cd in waters, macrozoobenthos and fish from one of the most developed mining areas in Armenia. Furthermore, hydro-ecological and health effects were also assessed. The topic is of great interest, but the MS lacks novelty.

We highly appreciate the reviewer for the response on this descriptive manuscript! We want our paper to be understandable for the general community, so we have expanded the description about heavy metal pollution in a river basin and some explanations. Please find below our detailed responses and corresponding revisions.

Generally, a lot of data are reported, even if it seems to be written at full speed with little overall processing of the ideas stated by the authors. In particular, the main concern regards the M&M section. M&M should be described with sufficient detail to allow others to replicate and build on published results. The number of samples analysed was not declared. How many water, macroinvertebrates and fish samples did you collect?

We have expanded the description about sampling design in MS (lines 87-109 and 124-127), which is given bellow.

Sampling was done in 8 locations of the rivers in the Debed catchment basin as out-lined in Figure 1 and Table 1. Water samples for the analysis of HMs were taken with polythene bottles pre-washed with 20% nitric acid (HNO3) as well as distilled water in April, July, and September 2017 and preserved with concentrated HNO3. Overall, 24 water samples were collected from the study area.

Macrozoobenthos samples for HM analysis were gathered in April 2017 with a Surber sampler. Benthic macroinvertebrates for the quantitative and qualitative analyses of animals were collected with a Surber sampler in April, July, and September 2017. The samples for HM and biological analyses were gathered from 5 points in each investigated location and mixed to make one sample for each location. The animal samples for HM analysis were stored in cool boxes under low temperature conditions, while the samples for biological investigation were preserved with a formaldehyde solution. Totally, 40 macroinvertabrate samples (5 samples from each of 8 locations) or 8 mixed samples were taken for HM analysis and 120 benthos samples or 24 mixed samples for biological analysis.

Fish samples (Capoeta Capoeta Guldenstadt 1773) for HM analysis were collected in April 2017 using a backpack electrofisher and stored in cool boxes under low temperature conditions until the laboratory treatments of the samples. Fish were obtained only from 6 locations, since no one fish was registered in the river sites L-4 and A-5. 3–5 samples of Kura scraper (Capoeta Capoeta Guldenstadt 1773) were gathered from each location. Overall, 25 fish samples were obtained for HM analysis.

The fish samples were dissected into gills, liver, and muscles, and the tissue samples obtained from scrapers collected from each location were mixed to make one sample for each organ. Totally, 6 mixed samples from the 6 locations were obtained for each organ.

The second concern regards the macroinvertebrates and fish samples. In literature it is well known that the trophic role influences metal accumulation. What kind of macroinvertebrates/fish species were selected? The terms “macrozoobenthos” or “fish” are too generic.

We have inserted information about fish species analysed for HMs in MS (lines 104-106 and 109-111), which is given bellow.

Fish samples (Capoeta Capoeta Guldenstadt 1773) for HM analysis were collected in April 2017 using a backpack electrofisher and stored in cool boxes under low temperature conditions until the laboratory treatments of the samples. Kura scraper was selected for HM analysis because it is one of widely distributed edible fish species in the Debed basin (Arakelyan, 2020) and one of most consumed species by local fishermen.

We measured HMs in the total biomass of macrozoobenthos and assessed adverse effects on animals at community level.

The citations and references are not well written. The authors didn’t follow the authors guidelines.

The citations and references have been changed according to the authors guidelines of the Journal (Done).

The third concern regards the lack of data from sediment (one of the main sink of metals in rivers).

The investigated hydro-ecosystems are fast-flowing mountain rivers, and the investigated river locations were characterized by sediments composed of mainly large stones and sands. All these make these sediments unsuitable for HM analysis.

Finally, the manuscript is not readable and must be linguistically checked (in particular a lot of prepositions are missing).

MS has been edited by a native speaker (Done).

Reviewer 2 Report

Dear author,

the paper entitled "Heavy Metal Contamination in an Industrially Affected River Catchment Basin: Assessment, Effects and Mitigation" is very interesting to the reader given valuable information about toxic HM in a river environment affected from industriial and minining activities in Armenia. Only few corrections are suggested: (1) please give QA/QC of the HM analyses procedure and also in any reference material is used for the different samples. (2) stylistic and some grammatical mistakes on the lines:21 (was not were), 33 (accumulate not accumulative), 94, 149 (Mueller), 239, 275, 385 (Diptera-italics), 424, 508 (Mueller) must be taken into consideration.

Author Response

Dear author, the paper entitled "Heavy Metal Contamination in an Industrially Affected River Catchment Basin: Assessment, Effects and Mitigation" is very interesting to the reader given valuable information about toxic HM in a river environment affected from industrial and minining activities in Armenia. Only few corrections are suggested: (1) please give QA/QC of the HM analyses procedure and also in any reference material is used for the different samples. (2) stylistic and some grammatical mistakes on the lines:21 (was not were), 33 (accumulate not accumulative), 94, 149 (Mueller), 239, 275, 385 (Diptera-italics), 424, 508 (Mueller) must be taken into consideration.

We highly appreciate the reviewer for the positive response on the manuscript! We want our paper to be understandable for the general community, so we have expanded the description about heavy metal pollution in a river basin and some explanations. Please find below our detailed responses and corresponding revisions.

We have inserted information about QA/QC of the HM analyses in MS (lines 137-142), which is given bellow.

All chemicals used were of analytical grade. Deionized water was used for preparation of calibration standards and in the analyses. All glassware used were pre-washed with 10% HNO3, followed by rinsing with distilled water prior to use. To ensure that HM analyzers remained calibrated during the experiments, certified reference materials and certified standard solutions were analysed for water and biological samples.

Line 21. We are talking about a group of fish of the same species, therefore, we should use “were”.

Line 33. We have replaced “accumulative” with “accumulate”.

Lines 94, 149, 424, 508. We have checked the author name in original reference, however, it is “Muller” as given in the MS, not Mueller.

Lines 239, 275, 385. Since Diptera is order, and in biology, taxonomic levels above genus should not be written as italic. We mistakenly wrote other orders as italic, but they have been corrected.

Round 2

Reviewer 1 Report

Previously, I have reviewed the first version of this manuscript providing comments for improving its overall quality. Authors have paid attention to those comments and modified the manuscript. However, my main concern regards the analysis on macroinvertebrates, which are composed by a lot of taxon belonging to several functional feeding groups (FFG).

In the M&M section the authors stated: "the fixed macrozoobenthos samples for quantitative and qualitative analyses were separated from the substrate and subsequently identified microscopically to taxonomic levels. The animals of each taxonomic group were dried on a filter paper and then quantified and weighed to obtain the total quantity and dry mass of each taxon". However, in the results section they reported the metal accumulation for "macrozoobenthos". Why did you choose to pool all taxon? FFGs influence trace element accumulation. How about the accumulation by each taxon? Please, explain. 

Author Response

We thank you for your valuable comment. However, we were not able to measure HM concentrations in macrozoobenthos at species level, since benthic community in the investigated rivers especially those affected by mining activities is characterized by very low quantitative and qualitative parameters which cause difficulties to collect enough biomass for taxon from each site for accurate analysis. To do such analyses in river basins under investigation, we have to sample a long river section (instead of a site) characterized by similar biotopes to obtain enough biomass for useful results. Therefore, we measured the HM concentrations in the total biomass of benthic animals which is suitable for these rivers.

We have gone through the manuscript once again and fixed references and a few words to enhance clarity of presentation. Authors hope that it meets your satisfaction.